

# The relationship of female physical attractiveness to body fatness

Guanlin Wang[1], Kurosh Djafarian[2], Chima A. Egedigwe[3], Asmaa El Hamdouchi[4], Robert Ojiambo[5], Harris Ramuth[6], Sandra Johanna Wallner-Liebmann[7], Sonja Lackner[7], Adama Diouf[8], Justina Sauciuvenaite[9], Catherine Hambly[9], Lobke M. Vaanholt[9], Mark D. Faries[10] and John R. Speakman[1,9]

[1] State Key Laboratory of Molecular Developmental Biology, Institute of Genetics and Developmental Biology, Chinese Academy of Sciences, Beijing, China
[2] Department of Clinical Nutrition, Tehran University of Medical Sciences, Tehran, Iran
[3] Department of Biochemistry, Michael Okpara University of Agriculture, Umuahia, Abia State, Nigeria
[4] CNESTEN, Unité Mixte de Recherche Nutrition et Alimentation, CNESTEN-Université Ibn Tofail, Rabat, Morocco
[5] College of Health Science, School of Medicine, Medical Physiology Department, Moi University, Eldoret, Kenya
[6] Biochemistry Department, Central health Laboratory services, Ministry of Health and Quality of Life, Mauritius
[7] Center of Molecular Medicine, Institute of Pathophysiology and Immunology, Medical University Graz, Graz, Austria
[8] Laboratoire de Nutrition, Département de Biologie Animale, Faculté des Sciences et Techniques, Université Cheikh Anta Diop de Dakar, Dakar, Senegal
[9] Institute of Biological and Environmental Sciences, University of Aberdeen, Aberdeen, UK
[10] Stephen F. Austin State University, Nacogdoches, TX, USA

Corresponding author
John R. Speakman,
j.speakman@genetics.ac.cn,
j.speakman@abdn.ac.uk

## ABSTRACT

Aspects of the female body may be attractive because they signal evolutionary fitness. Greater body fatness might reflect greater potential to survive famines, but individuals carrying larger fat stores may have poor health and lower fertility in non-famine conditions. A mathematical statistical model using epidemiological data linking fatness to fitness traits, predicted a peaked relationship between fatness and attractiveness (maximum at body mass index (BMI) = 22.8 to 24.8 depending on ethnicity and assumptions). Participants from three Caucasian populations (Austria, Lithuania and the UK), three Asian populations (China, Iran and Mauritius) and four African populations (Kenya, Morocco, Nigeria and Senegal) rated attractiveness of a series of female images varying in fatness (BMI) and waist to hip ratio (WHR). There was an inverse linear relationship between physical attractiveness and body fatness or BMI in all populations. Lower body fat was more attractive, down to at least BMI = 19. There was no peak in the relationship over the range we studied in any population. WHR was a significant independent but less important factor, which was more important (greater $r^2$) in African populations. Predictions based on the fitness model were not supported. Raters appeared to use body fat percentage (BF%) and BMI as markers of age. The covariance of BF% and BMI with age indicates that the role of body fatness alone, as a marker of attractiveness, has been overestimated.

## INTRODUCTION

Mate selection is a key behavioral component of reproduction related to the survival of ones' genes in the future gene pool (*Andersson & Simmons, 2006*; *Trivers, 1985*). Our perceptions of attractiveness of potential mates is complex and multi-dimensional, and may include many diverse aspects. These include economic parameters, like possessions, wealth and social economic status (SES) (*Drury, 2000*; *Swami et al., 2010*), psychological components such as cognitive ability, behavior, personality and social competence (*Eagly et al., 1991*), physiological aspects such as the major histocompatibility complex status (*Thornhill et al., 2003*), hormone levels (*Pawlowski & Sorokowski, 2008*) and age (*Borgerhoff Mulder, 1998*). In addition, physical aspects such as leg length (*Swami, Einon & Furnham, 2006b*), the shape of the face (*Grammer & Thornhill, 1994*; *Perrett et al., 1998*) and shape of the body (*Fallon & Rozin, 1985*; *Furnham, Tan & McManus, 1997*; *Singh, 1993*; *Singh & Young, 1995*; *Swami et al., 2006a*; *Swami & Tovee, 2005*; *Tovée et al., 2006*; *Tovee & Cornelissen, 2001*; *Tovee et al., 2002*; *Wass et al., 1997*) including the role of symmetry (*Perrett et al., 1998*; *Singh, 1993*; *Singh & Young, 1995*; *Smith, Cornelissen & Tovée, 2007*; *Tovee & Cornelissen, 2001*; *Tovee et al., 2002*) are also significant factors affecting attractiveness. The relative importance of these different dimensions for physical attractiveness may vary between the sexes and across cultures.

One aspect of attractiveness that has received considerable previous attention is the factors that drive perceptions of attractiveness of the female body (non-facial). Early studies focused on waist to hip ratio (WHR) (*Singh, 1993*; *Tovee et al., 2002*). A suggested preference for an optimal WHR around 0.7 has been generally interpreted within an evolutionary context because higher values of WHR are related to elevated risks of cardiovascular disease (*Terry, Page & Haskell, 1992*), diabetes (*Chan et al., 1994*) and cancer (*Borugian et al., 2003*). However, *Lassek & Gaulin (2008)* have suggested that WHR is not associated with health but more related to cognitive abilities. It has been noted, however, that WHR is not independent of body fatness, which may itself be an indicator of physical attractiveness. More recent work therefore has attempted to partition the importance of these two factors, and it has been conclusively shown across numerous studies that variation in attractiveness is much more closely related to variation in body fatness than to differences in WHR (e.g., *Henss, 2000*; *Kościński, 2013*; *Smith, Cornelissen & Tovée, 2007*; *Tassinary & Hansen, 1998*; *Tovee & Cornelissen, 1999*; *Tovee et al., 2002*; *Tovee & Cornelissen, 1999*; *Tovee et al., 1997*; *Tovee et al., 1998*).

Although many previous studies have set their observations into a *post hoc* evolutionary rationalization (e.g., *Borugian et al., 2003*), few studies have attempted to predict *a priori* the impact of different levels of body fatness based on an evolutionary model. Recent large scale epidemiological studies, linking variation in body fatness with risks of disease and fertility, and mathematical models that enable modeling of the relationship between fatness and famine survival, provide an opportunity to model much more closely the expected shape of the relationship between fatness and evolutionary fitness, and hence test whether physical attractiveness is indeed a marker of fitness. We argue that if physical attractiveness is related to fitness, then the relationship between body fatness and physical

attractiveness should mirror the relationship between fatness and fitness. Our aim was to develop such a model for the role of body fatness in physical attractiveness using, where available, culture specific data and then test the model across a range of different cultures, using a common protocol. We sampled independent populations drawn from the 3 dominant racial groups on earth: three Caucasian populations (Austria, Lithuania and UK), three Asian populations (China, Iran and Mauritius) and four African populations (Kenya, Morocco, Nigeria and Senegal). Caucasian, Africans and Asians together represent 91.4% of the current total world population (http://www.geohive.com/earth/world1.aspx). We found that females with lower body fatness (BMI and BF%) were rated as more attractive in all societies. WHR was also a significant factor that was more important (greater $r^2$) in African populations. Deviations from the evolutionary model were probably because raters used BMI as a proxy for subject's age.

## METHODS

### Evolutionary model

We considered that variation in female body fatness might have important fitness consequences for three different reasons: risk of fatal disease, impacts on fecundity, and survival under famine conditions. We searched the literature for epidemiological studies which had related the risks of mortality due to various individual fatal diseases, and all-cause mortality, to individual differences in body fatness. In addition we also sought studies that had linked together variations in body fatness and fertility. The thrifty gene hypothesis, first developed in the 1960s (*Neel, 1962*) with respect to diabetes, and subsequently elaborated in the context of obesity (*Eknoyan, 2001*; *Lev-Ran, 2001*), suggests that we have a genetic predisposition to obesity because in our evolutionary history we were regularly exposed to periods of famine. Individuals carrying 'thrifty genes', favouring the efficient deposition of fat reserves in the intervals between famines, would therefore be selected because they would have a greater chance of surviving the next famine. Body fatness is therefore an advantageous trait with respect to famine survival (but see *Speakman, 2007*; *Speakman, 2008*). Interestingly, in this context males who are more hungry alter their ratings of female attractiveness towards fatter subjects (*Swami & Tovee, 2006*). The exact relationship between survival in the absence of food and body fatness has been the subject of several mathematical models (*Hall, 2012*; *Song & Thomas, 2007*; *Speakman & Westerterp, 2013*). We used the outputs of such models to predict the shape of the relationship between body fatness and famine survival, and hence mortality risk (1/survival). We then combined these different impacts of body fatness on mortality, to produce two anticipated relationships between fatness and fitness: one including the effects of famine and one excluding such effects.

### Female body images

We used a series of 21 soft tissue dual-energy X-ray absorptiometry (DXA) images. DXA is a technique for evaluating body composition using the fact that bone, fat and lean tissue differentially absorb X-rays at different frequencies. The images we used had been

previously used (*Faries & Bartholomew, 2012*) to study the role of fatness in the perception of physical attractiveness in US college students. The 21 female body images covered 7 levels of body fat percentage (BF %): 15%–20%, 21%–25%, 26%–30%, 31%–35%, 36%–40%,41%–45%, and 46%–50%. At each level of BF% there were 3 levels of WHR: low (0.60–0.66), mid (0.67–0.75), and high (0.76–0.88) respectively. The BMI ranged from 19 to 40 kg m$^{-2}$. These images were selected from a database of over 5,000 female images and were specifically selected to break any correlation of BF% to WHR (*Faries & Bartholomew, 2012*). There was consequently a non-significant correlation between the two variables in these images ($r^2 = 0.029$, $p > .05$, for full details of the images and their characteristics see (*Faries & Bartholomew, 2012*). We could not use a wider range of body fatness because it was not possible to find images with higher or lower BMI with also the desired range of WHR. The range we used spanned all the predicted peaks in the relationship between BMI and fitness derived from the evolutionary models.

Ages of the subjects in the images were known, but not controlled for, or revealed to the raters. By using DXA images the facial details were not a factor influencing the subjects' judgment of attractiveness. Two of the images showing a constant level of WHR at two very different levels of body fatness are shown in (Fig. S1). For the present study, the images were printed on an A3 sheet of heavy paper and the individual images were then cut out into playing card sized rectangles. The number of the image (1 to 21) (*Faries & Bartholomew, 2012*) was written on the back of each card.

## Participants

Participants ($N = 1,327$ in total) were recruited from major cities in ten countries: Graz in Austria; Panevezys in Lithuania; Aberdeen in the UK; Beijing in China; Tehran in Iran; Port Louis in Mauritius; Eldoret in Kenya; Tiflet, Kenitra, Casablanca, Rabat and Oujda in Morocco; Umuahia, Abia state, in Nigeria; and Dakar in Senegal (Table 1). Participation was voluntary and verbal informed consent was obtained before the study. All the procedures for the overall study were ethically reviewed and approved by the Chinese Academy of Sciences,Institute of Genetics and Developmental Biology Institutional Review Board (IGDB-2013-IRB-005). In addition, local ethical approval was also obtained at the UK site from the University of Aberdeen College of Life Science and Medicine Ethical Review Board (CERB/2014/12/1123).

## Procedure

Participants (raters) were asked for some basic information (age, sex, ethnicity, height, weight) before the task started. They were then given the 21 image cards which were shuffled and placed on a table in front of them in a random order. Participants were then asked to reorder the cards from the most attractive on their right to the least attractive on their left. They were not allowed to have ties. The sorting task took about 5 min to complete.

The recorder then recorded the order of the images and confirmed with the subject that the order was indeed from least to most attractive and not the reverse. We predominantly selected subjects in the age range 18 to 50, except in Mauritius where the subjects were
**Table 1** Details of the rating participants from each country.

| Country | Sample size | | | Age (mean ± S.D.) | | | BMI (mean ± S.D.) | | |
|---|---|---|---|---|---|---|---|---|---|
| | N | Female | Male | All | Female | Male | All | Female | Male |
| Austria | 53 | 45 | 8 | 27.5 ± 9.8 | 26.8 ± 8.8 | 31.5 ± 14.2 | 24.0 ± 5.9 | 23.8 ± 6.1 | 24.9 ± 4.3 |
| UK | 85 | 48 | 37 | 23.2 ± 6.1 | 24.1 ± 7.1 | 22.0 ± 4.3 | 23.1 ± 3.7 | 22.6 ± 3.9 | 23.6 ± 3.5 |
| Lithuania | 60 | 41 | 19 | 34.1 ± 11.9 | 36.6 ± 12.1 | 28.7 ± 9.8 | 23.9 ± 3.9 | 24.1 ± 4.1 | 23.5 ± 3.7 |
| China | 209 | 98 | 111 | 25.4 ± 5.3 | 25.7 ± 6.1 | 25.2 ± 4.5 | 21.5 ± 2.5 | 20.5 ± 2.2 | 22.3 ± 2.6 |
| Iran | 180 | 115 | 65 | 30.2 ± 10.6 | 31.0 ± 10.4 | 28.8 ± 10.9 | 26.8 ± 6.3 | 27.0 ± 5.8 | 26.4 ± 7.2 |
| Mauritius | 62 | 44 | 18 | 13.5 ± 1.7 | 13.6 ± 1.9 | 13.2 ± 1.2 | 19.9 ± 5.4 | 20.1 ± 5.6 | 19.4 ± 5.1 |
| Nigeria | 179 | 116 | 62 | 27.6 ± 8.1 | 27.4 ± 7.7 | 27.8 ± 8.5 | 23.8 ± 4.2 | 23.7 ± 4.4 | 23.9 ± 3.9 |
| Kenya | 104 | 43 | 61 | 22.3 ± 4.1 | 21.0 ± 1.7 | 23.2 ± 5.0 | 21.8 ± 2.9 | 22.3 ± 2.8 | 21.4 ± 2.9 |
| Morocco | 260 | 132 | 128 | 24.1 ± 4.8 | 23.5 ± 3.9 | 24.7 ± 5.5 | 22.9 ± 3.0 | 22.5 ± 3.2 | 23.3 ± 2.8 |
| Senegal | 135 | 135 | 0 | 25.3 ± 3.9 | 25.3 ± 3.9 | — | 22.7 ± 5.8 | 22.7 ± 5.8 | — |
| **Total**[a] | **1327** | **817** | **509** | | | | | | |

**Notes.**
[a] One missing gender in Nigeria population.

adolescents (Table 1). The populations varied significantly in their mean BMI (Table 1). We did not exclude anyone according to their sexual orientation. Homosexuality is illegal or highly stigmatized in several of the countries involved in the study and we therefore did not consider that self reports of sexual orientation would be reliable. We also did not control rater hunger or stress levels both of which have been previously implicated as influencing ratings of female attractiveness (*Swami & Tovee, 2006*; *Swami & Tovee, 2012*).

The methodology used here differed slightly from that in *Faries & Bartholomew (2012)*. In that study individuals were asked to select their most and least attractive images from the set and give them ratings of 9 and 1 respectively and then use these anchors to grade all the other images on a scale from 1 to 9. In this process it was possible to get ties. To evaluate whether the resultant ratings were similar across the two methodologies we also applied this procedure to the subjects in the UK. There was a very strong correlation ($r^2 = 0.95$) in the ratings of the images between the two protocols lending confidence to the fact we could directly compare our data to those collected previously in the USA, despite the slight protocol difference.

## Standard score

The rank positions of the images were converted to a score in the range 1 to 9. The score followed the formula $a_n = 1 + (n - 1) * 0.4$ (where n was the rank order of the image from the least attractive to the most attractive i.e., n of the least attractive image was 1 so the score was $a_1 = 1 + (1-1) * 0.4 = 1$ and the most attractive image was 21 so the score was $a_{21} = 1 + (21-1) * 0.4 = 9$)

## Age ratings

Participants ($N = 325$, from Austria, China, Iran, Kenya, Morocco and Senegal) took part in this task. Raters were asked for some basic information (age, sex, ethnicity, height,

weight) before the task started. They were given the same 21 images on an A4 paper with a separate list of the actual ages of the subjects in the images (21 images with 21 ages). They were asked to match together the age and the image. The task took about 5 min to complete.

## Statistical analysis

Software including R, SPSS 11.5 and Minitab 16 were used to analyze data. Pearson correlation was used to explore the overall correlation of the rankings between the sexes of the raters. In addition we compared the ratings of female and male raters for all the images individually in each country (corrected for multiple testing in each country using the Bonferroni correction) using the non-parametric Mann Whitney test as data were often not normally distributed. We performed univariate analyses using mean attractiveness across all the raters and %BF, BMI or WHR as predictors in least squares linear regression analysis for each country separately. We then performed analyses using general linear modeling with mean attractiveness across all raters as the dependent variable, and the picture BF%, BF%-squared, WHR and age and all the two way interactions as fixed factors. Analyses were conducted separately for each country. Mathematical modeling and analysis was performed in Mathcad 15.

# RESULTS

## Evolutionary model of the relationship between attractiveness and fatness

Many studies have associated the risk of developing various diseases with different levels of body fatness (*Borugian et al., 2003*; *Chan et al., 1994*; *Despres, 2012*; *Terry, Page & Haskell, 1992*). We found three reviews which compiled data for different ethnic groups to establish ethnic specific patterns of mortality in relation to fatness. These included reviews involving >900 k Caucasians (*Whitlock et al., 2009*) and >1.1 million Asians (*Zheng et al., 2011*). We could not locate any summary of the same relationship pertaining to Africans living in Africa, but found reviews including >360 k African Americans (*Cohen et al., 2014*; *Cohen et al., 2012*; *Flegal et al., 2013*). In the studies involving Caucasians and African Americans the data was subdivided by gender so we could extract female specific curves, but for the Asians this was not possible from the data in the original paper. However, the patterns for males and females in the Caucasians and African Americans were almost identical so this is unlikely to be a serious source of error. The pattern of all cause mortality (total mortality irrespective of cause) from these three studies in relation to BMI is shown in Fig. 1A. We expressed the mortality in each BMI class as the excess mortality above that of the lowest BMI class, since this reflects the negative impact of differences in body fatness, and then fitted a polynomial to the data for each ethnic group using ordinary least squares regression. The resultant best fit (least squares) equations were a series of third order polynomials which explained respectively 97.1% for Caucasians (Eq. (1a)), 99.8% for Asians (Eq. (1b)) and 98.4% for African Americans (Eq. (1c)), of the variance in excess

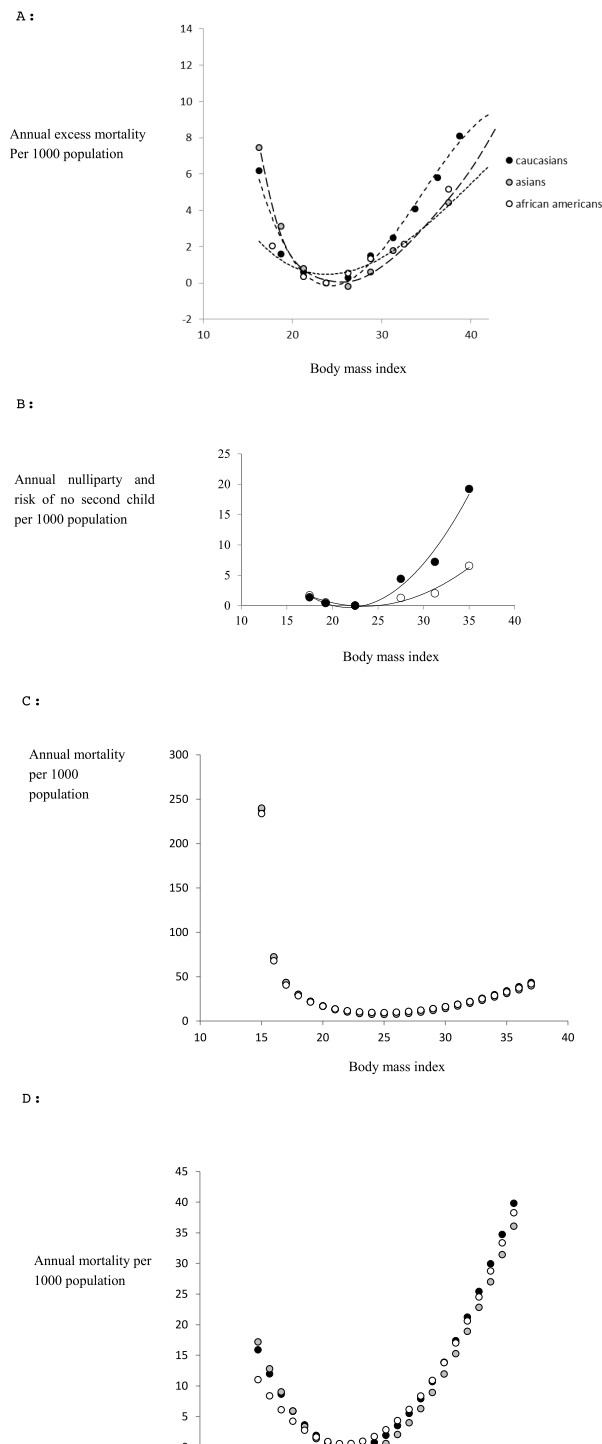

**Figure 1 Evolutionary model.** (A) Epidemiological data linking all cause mortality to body fatness (BMI) for female subjects minus the mortality for the class with the lowest mortality (data from *Whitlock et al., 2009* for Caucasians; (*Zheng et al., 2011*) for Asians and *Cohen et al., 2012*; *Cohen et al., 2014* for African Americans). The curves are the fitted third order polynomials (continued on next page...)

**Figure 1 (...continued)**
(see text for details) (B) probability of nulliparity over entire reproductive age annualized per 1,000 population (open symbols) and probability of not having a second child if one child has already been born annualized per 1,000 population (closed symbols) as a function of BMI at age 20. Data are subtracted from the class with the lowest probabilities (data from *Jacobsen et al. (2013)*). (C) Combined effects of infertility and all cause mortality in relation to BMI (effective mortality risk per 1,000 population) for each ethnic group. The minimum point of the curve is at BMI $= 23.18$ for Caucasians, 23.12 for Asians and 22.45 for African Americans (see text for derivation details). (D) Combined effects of infertility and all cause mortality (as in c) as well as the impact of fatness on famine survival on the relationship between mortality and Body mass index (effective mortality per 1,000 population). The minimum points of the curves are at BMI $= 24.78$ for Caucasians, 24.72 for Asians and 24.05 for African Americans (see text for derivation details).

annual mortality

$$y_{1c} = -0.002359x^3 + 0.24392x^2 - 7.6714x + 76.089 \tag{1a}$$

$$y_{1a} = -0.0034x^3 + 0.3286x^2 - 10.004x + 97.859 \tag{1b}$$

$$y_{1aa} = -0.0005x^3 + 0.0649x^2 - 2.2071x + 23.272 \tag{1c}$$

where $y_{1c}, y_{1a}, y_{1aa}$ are the excess annual mortalities per thousand population due to all causes and x is the BMI for Caucasian, Asian and African American populations respectively. Many studies have also studied aspects of reproductive biology in relation to body fatness (or BMI). However, we could not find any summaries for Asian or African/African American populations. Among the most comprehensive studies of Caucasians was the Adventist Health study (*Jacobsen et al., 2013*) which included lifetime fertility records for 33,159 females along with their BMI at age 20. The relationship between the probability of having no children during a reproductive life of 20 years, and BMI class at age 20 is shown in Fig. 1B. The 20 year excess probability of not having children, compared to the BMI class with the lowest rate of nulliparity, contributes to the negative effect of BMI on fertility. To obtain the annualized rate of excess 'missing births' we divided this lifetime rate by 20 and then fitted a polynomial to these data using ordinary least squares regression. In this case the best fit was a second order polynomial which explained 97.5% of the variation

$$y_2 = 0.1065x^2 - 4.6346x + 50.145 \tag{2}$$

The same study also showed that the probability of having a second child was also impacted by obesity status at age 20. The data are also shown in Fig. 1B and in this case the excess missing births relative to the BMI class with the lowest rate of not having a second child were best described by a second order polynomial

$$y_3 = 0.0478x^2 - 2.2438x + 26.282 \tag{3}$$

which explained 95.9% of the variation.

Given the similarity in the relationships between mortality and BMI among the different ethnic populations we assumed that the relationships between BMI and fecundity for all
ethnic groups were adequately represented by these Caucasian data. Hence the combined effects of fatness on all cause mortality and reduced fecundity can be expressed as

$$y_{\text{total}} = y_1 + y_2 + y_3. \tag{4}$$

Substituting from Eqs. (1) to (3) into (4), collecting terms and simplifying yields three ethnic specific relationships for Caucasians, Asians and African Americans.

$$y_{\text{totalc}} = -0.00236x^3 + 0.3982x^2 - 14.550x + 152.52 \tag{5a}$$
$$y_{\text{totala}} = -0.00034x^3 + 0.4829x^2 - 16.8824x + 171.286 \tag{5b}$$
$$y_{\text{totalaa}} = -0.0005x^3 + 0.2192x^2 - 9.0855x + 99.70. \tag{5c}$$

These composite curves expressing total excess mortality and reduced fecundity are shown in Fig. 1C. Differentiating Eqs. (5a) to (5c) gives

$$\frac{dy}{dx} = -0.00708x^2 + 0.7964x - 14.5498 \tag{6a}$$

$$\frac{dy}{dx} = -0.0102x^2 + 0.9658x - 16.8824 \tag{6b}$$

$$\frac{dy}{dx} = -0.0015x^2 + 0.4384x - 9.0855. \tag{6c}$$

We can then solve these quadratic Eqs. (6a) to (6c) $f(x) = 0$ to obtain the BMI at the minimum point and this yields

$$0 = 0.00708(x - 88.673)(x - 23.182) \tag{7a}$$
$$0 = 0.0102(x - 71.554)(x - 23.131) \tag{7b}$$
$$0 = 0.0015(x - 269.818)(x - 22.448). \tag{7c}$$

Each of which has only one solution in the range BMI 15 to 40 which is 23.182 for Caucasians, 23.131 for Asians and 22.448 for African Americans. Therefore the BMI at the peak of the fitness function (lowest mortality) was between 22.448 and 23.182 depending on ethnicity. This model predicts therefore that if attractiveness is directly related to fitness, combining future potential fertility with all cause mortality, the relationship between attractiveness and BMI should have a peaked function, with the maximum attractiveness at a BMI around 22.4 to 23.2. Given the shape of the function in Fig. 1C we would expect the attractiveness function to be similarly distributed about this peak.

During famine all mortality may be considered 'excess mortality'. We previously (*Speakman & Westerterp, 2013*) constructed a mathematical model of energy utilization during complete starvation to predict the survival durations of people at different starting body fatness (or BMI). This model is not dependent on the race of the individual. For females the survival function was

$$\text{Survival (days)} = 12.306x - 180.1. \tag{8}$$

Since mortality risk is the inverse of survival duration we can express the excess mortality per thousand population as

$$y_4 = \frac{1,000}{(12.306x - 180.1)}. \tag{9}$$

And hence adding $y_4$ to the ethnic specific estimates of $y_{total}$ gives the total estimated mortality risk including famine mortality for each race (Caucasian: 10.1, Asians 10.2 and African Americans 10.3) as

$$y_{totalc} = -0.002359x^3 + 0.39822x^2 - 14.5498x + 152.516 + \frac{1,000}{(12.306x - 180.1)} \tag{10a}$$

$$y_{totala} = -0.00034x^3 + 0.4829x^2 - 16.8824x + 171.286 + \frac{1,000}{(12.306x - 180.1)} \tag{10b}$$

$$y_{totalaa} = -0.0005x^3 + 0.2192x^2 - 9.0855x + 99.70 + \frac{1,000}{(12.306x - 180.1)}. \tag{10c}$$

The curve relating mortality to BMI represented by Eqs. (10a) to (10c) are shown in Fig. 1D. Differentiating Eqs. (10a) to (10c) yields

$$\frac{dy}{dx} = -0.00708x^2 - 0.7964x - 14.9498 + \frac{12,306}{(12.306x - 180.1)^2} \tag{11a}$$

$$\frac{dy}{dx} = -0.0102x^2 + 0.9658x - 16.8824 + \frac{12,306}{(12.306x - 180.1)^2} \tag{11b}$$

$$\frac{dy}{dx} = -0.0015x^2 + 0.4384x - 9.0855 + \frac{12,306}{(12.306x - 180.1)^2} \tag{11c}$$

and solving Eq. (11) for $f'(x) = 0$ gives a single root for each ethnic group: Caucasians (Eq. (11a)) $x = 24.78$, Asians (Eq. (11b)) $x = 24.72$ and African Americans (Eq. (11c)) $x = 24.05$. Hence including mortality due to famine into the prediction shifts the peak upwards and strongly accentuates the negative aspects of being leaner than this optimum. If attractiveness is related to fitness the actual curve relating attractiveness to body fatness might be expected to lie somewhere between the inverses of the curves depicted in Fig. 1C and 1D, depending on the perceived risk in a given population that there will be a famine. This might for example depend on the duration since the last famine occurred in a given population.

## Comparison of ratings of female physical attractiveness by males and females

To investigate the influence of rater sex on perceived attractiveness we performed the study using both sexes as raters. Scatter plots of the average attractiveness rating of females raters against male raters for the 21 images in each of the nine populations [excluding Senegal where all the participants were female] showed that there was strong concordance in the perceptions of female attractiveness between the sexes in all populations (UK: $R^2 = 0.9778$; China: $R^2 = 0.99$; Iran: $R^2 = 0.9888$; Mauritius, $R^2 = 0.97$; Kenya: $R^2 = 0.9791$; Morocco $R^2 = 0.9906$; Nigeria: $R^2 = 0.9428$) (Fig. 2). We also explored whether individual images

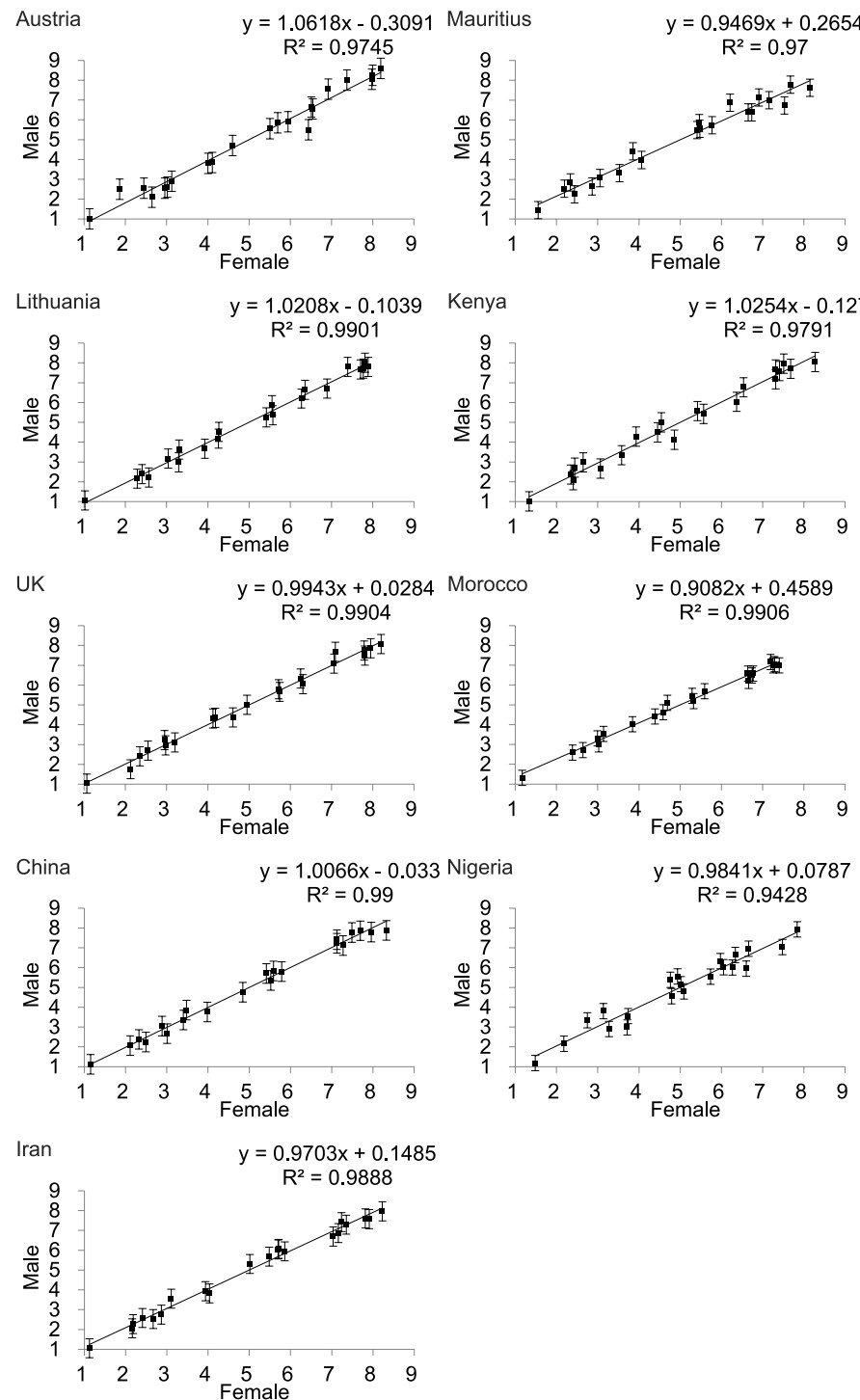

**Figure 2** Relationship between the rankings by males and females of the attractiveness of 21 DXA soft tissue images of females, of varying BMI and waist to hip ratio, across 9 populations (except Senegal). The *X*-axis is the rating by females and the *Y*-axis the rating by males.

were rated differently. There were no significant differences between female and male raters at the 95% confidence level (Bonferroni corrected for 188 tests) for all of the images across nine countries (Table S1). Although there was no overall effect when using the Bonferroni correction, we noted an interesting pattern in the unadjusted probability values. Among the Caucasian populations only 1/62 tests showed a significant difference in the ratings between males and females. In the Asian populations 9/63 tests were significant and in the African populations 14/63 tests showed significantly different ratings between males and females. Hence, while there was no overall effect using the adjusted values, it is possible there were ethnic differences in the extent to which males and females agreed on the attractiveness of particular images, and the use of the Bonferroni correction was too stringent to allow us to detect this effect.

## Effects of BF% and WHR: univariate analyses

Scatter plots of female attractiveness in relation to BF% (Fig. 3) BMI (Fig. S2) and WHR (Fig. 4) were generated for each country and univariate analyses performed. We found a significant negative linear relationship between BF% and attractiveness in all the populations. Parameters of the univariate regression models are in Table 2. Across all the populations the linear fit models explained between 46.3 and 85.3% of the variance in attractiveness. The poorest fits were for Nigeria and Senegal. Including BF% squared did not result in a significant improvement in any of the relationships. The pattern for BMI was almost identical (Fig. S2). In none of the populations did a peaked relationship fit the data, contrary to what was predicted *a priori* from the evolutionary model (Fig. 1).

WHR also showed a linear relationship to attractiveness rating (Fig. 4) but in this instance the fits were much poorer than for BF% or BMI (Table 2) and for all three of the Asian countries WHR was not significantly associated with attractiveness ($p > .05$).

## Multiple regression analyses

None of the two way interactions between BF%, WHR and age were significant and these terms were removed from the final models. When BF%, WHR and age were included as independent predictors they all entered as significant predictors ($P < .01$) in all countries except Mauritius ($P = 0.028$: not significant at $P = .05$ when corrected for multiple testing). This occurred even though in the other Asian countries WHR had not been significant in the univariate analysis (above). In Mauritius only BF% was a significant predictor. BF%-squared was not a significant term in any of the models. Age of the subjects in the images was also a significant predictor ($P < 0.05$) except in Austria ($P = 0.129$) and UK ($P = 0.059$) (Table 3), despite the images containing no overt indication of the subject age. Parameters of the fitted multiple regression models are presented in Table 3.

## Age relationship to BF% and BMI

There was no significant correlation between the estimated age of the figures and their actual ages across all 6 populations involved in this part of the study (Fig. 5A). However, there was a strong positive relationship between estimated age and BF% ($r^2 = 0.812$)

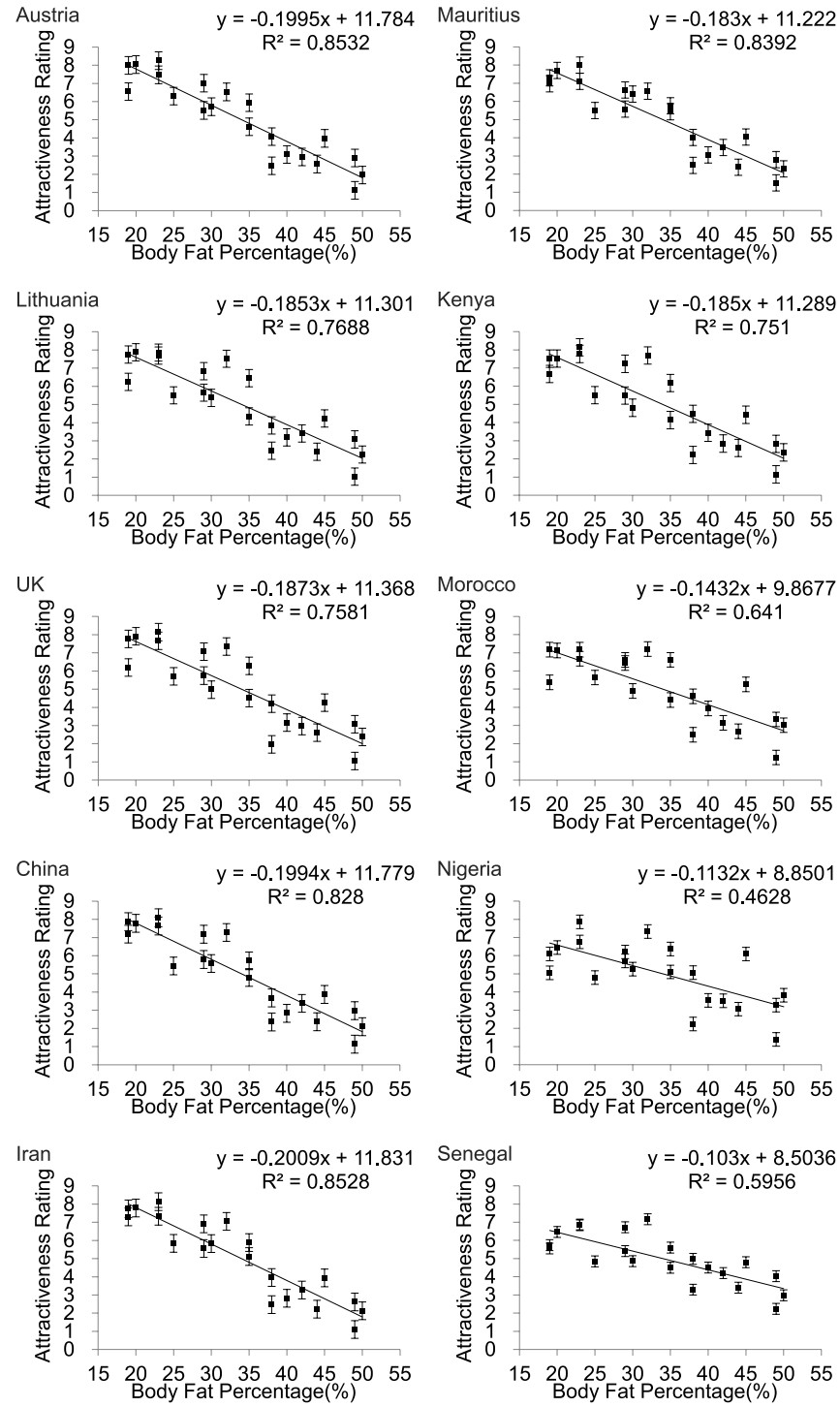

**Figure 3 Body fat percentage to attractiveness.** Relationships between the average ratings of physical attractiveness of 21 DXA soft tissue images and body fat % of the subjects in the images across ten different populations. Error bar referred to the standard error of both directions.

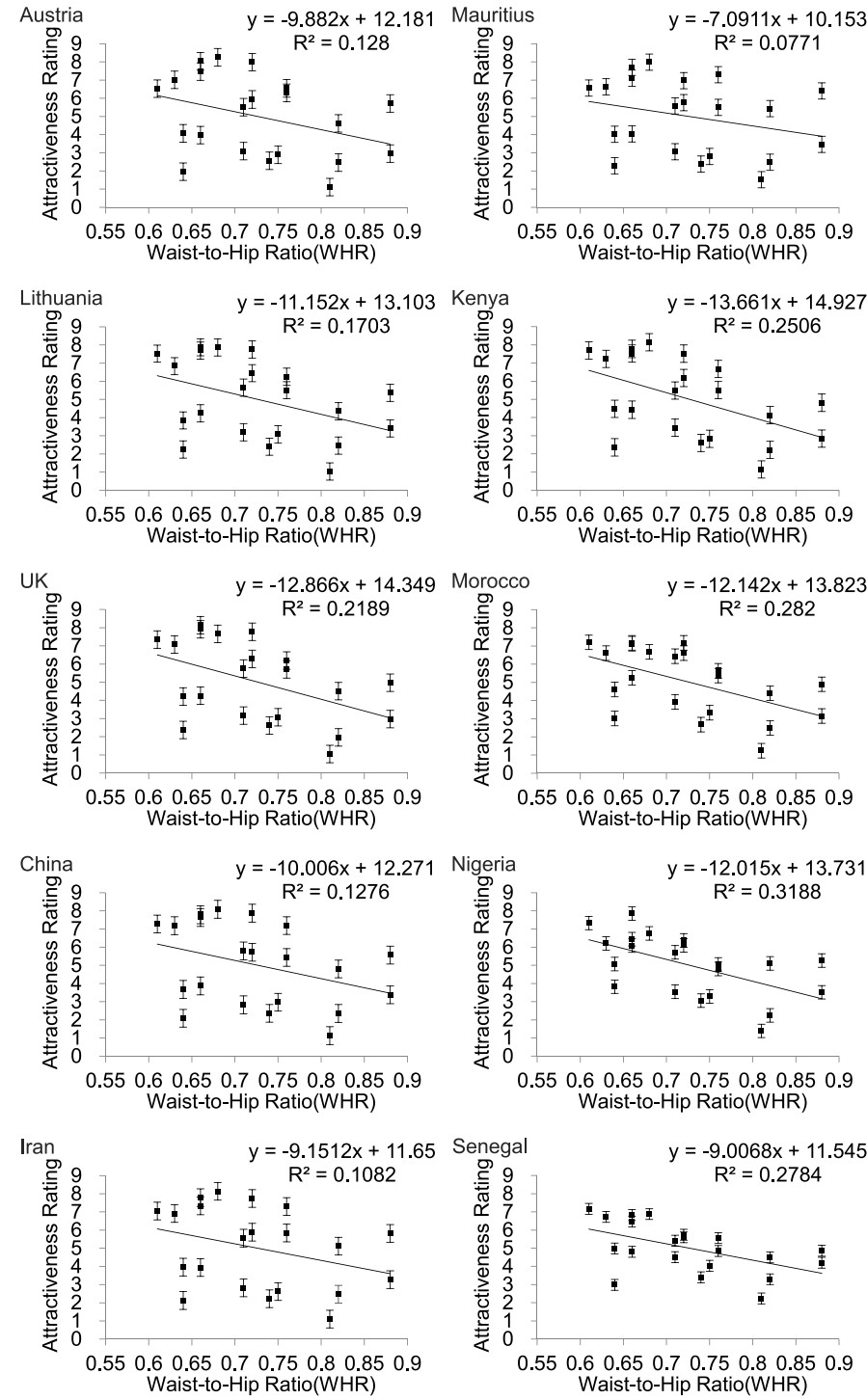

**Figure 4** Relationships between the average ratings of physical attractiveness of 21 DXA soft tissue images and waist to hip ratios (WHR) of the subjects in the images across ten different populations. Error bar referred to the standard error of both directions.

**Table 2  Univariate analyses.** Parameters of least squares fit regression equations relating BF%, BMI and WHR to average attractiveness across 21 DXA soft tissue images. In all cases the df for the $F$ statistic was 1,19.

| Country | Equation | $R$ square | $F$ | $P$-value |
|---|---|---|---|---|
| **(A) BF%** | | | | |
| Austria | $y = -0.1995x + 11.784$ | 0.853 | 110.41 | <0.001 |
| Lithuania | $y = -0.185x + 11.301$ | 0.769 | 63.18 | <0.001 |
| UK | $y = -0.187x + 11.368$ | 0.758 | 59.55 | <0.001 |
| China | $y = -0.199x + 11.779$ | 0.828 | 91.48 | <0.001 |
| Iran | $y = -0.201x + 11.831$ | 0.853 | 110.07 | <0.001 |
| Mauritius | $y = -0.183x + 11.222$ | 0.839 | 99.18 | <0.001 |
| Kenya | $y = -0.185x + 11.289$ | 0.751 | 57.3 | <0.001 |
| Morocco | $y = -0.143x + 9.868$ | 0.641 | 33.92 | <0.001 |
| Nigeria | $y = -0.113x + 8.850$ | 0.463 | 16.37 | <0.001 |
| Senegal | $y = -0.103x + 8.504$ | 0.596 | 27.99 | <0.001 |
| **(B) BMI** | | | | |
| Austria | $y = -0.3833 + 14.766$ | 0.767 | 62.39 | <0.001 |
| Lithuania | $y = -0.369x + 14.397$ | 0.741 | 54.51 | <0.001 |
| UK | $y = -0.376x + 14.574$ | 0.743 | 54.94 | <0.001 |
| China | $y = -0.39x + 14.941$ | 0.772 | 64.36 | <0.001 |
| Iran | $y = -0.394x + 15.03$ | 0.797 | 74.66 | <0.001 |
| Mauritius | $y = -0.360x + 14.164$ | 0.789 | 71.19 | <0.001 |
| Kenya | $y = -0.376x + 14.586$ | 0.756 | 59.04 | <0.001 |
| Morocco | $y = -0.301x + 12.67$ | 0.69 | 42.29 | <0.001 |
| Nigeria | $y = -0.279x + 12.095$ | 0.682 | 40.66 | <0.001 |
| Senegal | $y = -0.232x + 10.897$ | 0.732 | 51.80 | <0.001 |
| **(C) WHR** | | | | |
| Austria | $y = -9.882x + 12.181$ | 0.128 | 2.79 | >0.05 |
| Lithuania | $y = -11.152x + 13.103$ | 0.170 | 3.90 | >0.05 |
| UK | $y = -12.866x + 14.349$ | 0.219 | 5.32 | <0.05 |
| China | $y = -10x + 12.271$ | 0.128 | 2.78 | >0.05 |
| Iran | $y = -9.15x + 11.65$ | 0.108 | 2.31 | >0.05 |
| Mauritius | $y = -7.09x + 10.153$ | 0.077 | 1.59 | >0.05 |
| Kenya | $y = -13.66x + 14.927$ | 0.251 | 6.35 | <0.05 |
| Morocco | $y = -12.14x + 13.823$ | 0.282 | 7.46 | <0.05 |
| Nigeria | $y = -12.02x + 13.731$ | 0.319 | 8.89 | <0.01 |
| Senegal | $y = -9.007 + 11.545$ | 0.278 | 7 | <0.05 |

(Fig. 5B) and between estimated age and BMI ($r^2 = 0.848$) (Fig. 5C). There was also no significant relationship between estimated age and WHR (Fig. 5D).

## DISCUSSION

### Ratings by males v. females

Several previous studies have compared the ratings made by males and females of female attractiveness. Similar to the results from our study, in most previous studies it was
**Table 3 Multiple regression analyses.** Effects of subject body fatness (BF%), waist to hip ratio (WHR) and age on average attractiveness using general linear models run separately for each of seven separate populations. Parameters of the full models and regression coefficients are in Table S2. df for all $F$ statistics are 1,17.

| Population | Overall $r^2$ | Age | | WHR | | BF% | |
|---|---|---|---|---|---|---|---|
| | | $F$ | $P$ | $F$ | $P$ | $F$ | $P$ |
| Austria | 0.9139 | 2.54 | 0.129 | 11.59 | 0.003 | 111.28 | <0.001 |
| Lithuania | 0.8887 | 6.20 | 0.023 | 16.44 | 0.001 | 67.14 | <0.001 |
| UK | 0.8973 | 4.09 | 0.059 | 22.58 | <0.001 | 73.89 | <0.001 |
| China | 0.9101 | 6.34 | 0.022 | 13.19 | 0.002 | 95.05 | <0.001 |
| Iran | 0.9154 | 5.36 | 0.033 | 10.54 | 0.005 | 108.19 | <0.001 |
| Mauritius | 0.8954 | 6.01 | 0.025 | 5.81 | 0.028 | 84.8 | <0.001 |
| Kenya | 0.9089 | 7.26 | 0.015 | 36.71 | <0.001 | 92.58 | <0.001 |
| Morocco | 0.8686 | 7.36 | 0.015 | 27.95 | <0.001 | 40.69 | <0.001 |
| Nigeria | 0.7894 | 8.25 | 0.011 | 22.99 | <0.001 | 13.26 | 0.002 |
| Senegal | 0.8620 | 11.70 | 0.003 | 29.12 | <0.001 | 31.52 | <0.001 |

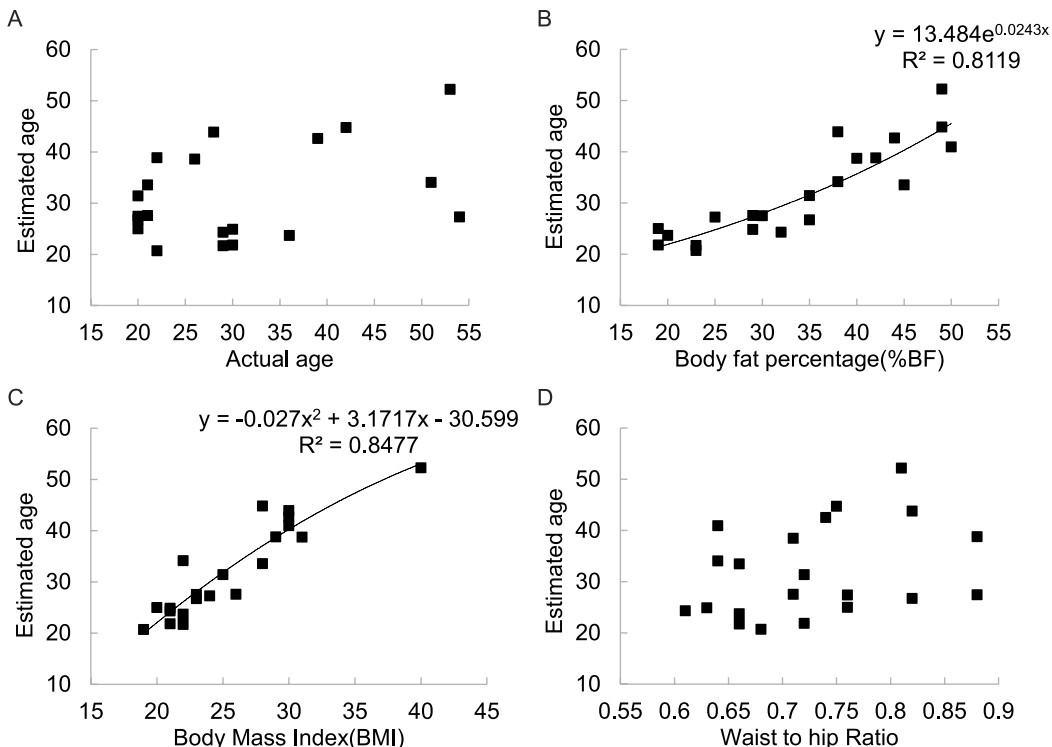

**Figure 5 Relationships between estimated subject age and (A) actual subject age, (B) subject body fatness, (C) subject BMI, (D) subject WHR for 21 DXA soft tissue images averaged across 325 mixed sex raters in six different countries.**

observed that males and females in a given population did not differ in their ratings of female attractiveness (*Faries & Bartholomew, 2012*; *Furnham, Tan & McManus, 1997*; *Henss, 1995*; *Henss, 2000*; *Kościński, 2013*; *Schmalt, 2006*; *Singh, 1994*; *Streeter & McBurney, 2003*; *Swami et al., 2006a*; *Swami & Tovee, 2005*; *Tassinary & Hansen, 1998*; *Tovée et al., 2006*; *Tovee & Cornelissen, 2001*; *Tovee et al., 2002*). This was despite sometimes large differences in what the different populations perceived as attractive (see in particular (*Tovée et al., 2006*)). In contrast some studies have found that females rated attractiveness differently from males with respect to WHR (*Furnham, Dias & McClelland, 1998*). The reasons for differences between studies are unclear. Perhaps it is evolutionarily advantageous for males and females to perceive attractiveness in their own populations in the same way, and most studies including ours indicate this is the case. However, we did note that males and females were more likely to make divergent estimates of attractiveness of given images in Asian and African populations relative to Caucasians. The reason for this ethnic difference is unclear.

## Body fatness v. waist to hip ratio

Early studies regarding the relationship between female body shape and physical attractiveness placed considerable emphasis on the role played by WHR (*Henss, 2000*; *Mo et al., 2014*; *Price et al., 2013*; *Singh, 1993*; *Singh, 1994*; *Singh, 1995b*; *Tassinary & Hansen, 1998*; *Tovee et al., 2002*). These studies suggested an optimal WHR of around 0.7 was maximally physically attractive (*Dixson et al., 2007*; *Marlowe & Wetsman, 2001*). This was consistent with the fact that Playboy centerfolds and glamour models almost all having WHRs between 0.6 and 0.7 (*Katzmarzyk & Davis, 2001*; *Voracek & Fisher, 2002*) and this changed only very slightly over the 5 decades from 1950 to 2000. Moreover, when individuals are asked to manipulate graphics to generate their ideal body shape, they routinely select a WHR of 0.7 (females) to 0.73 (males) (*Crossley, Cornelissen & Tovee, 2012*). Surgical enhancement of WHR increases physical ratings of attractiveness (*Dixson et al., 2007*). WHR was interpreted within an evolutionary context as of importance because it was suggested to be an honest signal of health and fertility (*Bigaard et al., 2004*; *Singh, 1995a*; *Wass et al., 1997*) or cognitive ability (Lassek and Gualin, 2008). An early suggestion that females with higher WHR might have more sons was later dismissed (*Tovee, Brown & Jacobs, 2001*).

However, it was pointed out in the late 1990s and early 2000s that WHR is not independent of body fatness, which could itself act as an honest signal of health and fertility (*Tovee & Cornelissen, 1999*; *Tovee et al., 1999*; *Tovee et al., 1998*) and hence be the primary signal indicating female attractiveness (*Tovee et al., 1999*; *Tovee et al., 1998*). Since this time a large number of studies have attempted to partition the variation in physical attractiveness that is explained by body fatness (or body mass index) and WHR. These studies showed conclusively, across many cultures, that variation in BMI was a better indicator of physical attractiveness than WHR (*Kościński, 2013*; *Swami et al., 2006a*; *Swami & Tovee, 2005*; *Tovée et al., 2006*; *Tovee & Cornelissen, 2001*; *Tovee et al., 2002*; *Tovee & Cornelissen, 1999*; *Tovee et al., 1998*). This was true even when the variation in the two traits

was held constant, (*Swami et al., 2006a*; *Swami & Tovee, 2005*; *Tovée et al., 2006*; *Tovee et al., 2002*) or the relationship between BMI and WHR was made artificially negative (*Tovee & Cornelissen, 1999*).

Our own study confirms this general pattern that body fatness explains more of the variation in physical attractiveness than does WHR. In contrast to many previous studies however, the relative importance of body fatness and WHR in our data was culturally dependent. In the univariate analysis the variance explained by body fatness was considerably greater in the Asian and Caucasian populations (75 to 85%) than in the four African populations (46 to 75%). Moreover, in the univariate analysis there was no significant effect of WHR in the three Asian populations, but in the African populations the explained variation by WHR alone was 25 to 32%. A previous study using the same images rated by a predominantly Caucasian population (70% Caucasian) in the USA found 70% of the variance was explained by body fatness and 18% by WHR (*Faries & Bartholomew, 2012*) consistent with the 76% and 21.9% respectively for the Caucasian UK sample and the 77% and 17% respectively in Lithuanians measured here. The very high percentage variation explained by body fatness (BMI) and the low variation explained by WHR in the Asian populations was consistent with the variation in attractiveness explained by BMI and WHR in Malaysians (*Swami & Tovee, 2005*) and Thai subjects (*Swami & Tovée, 2007*), but in Japanese subjects the variance explained by WHR was higher at 30.2% (*Swami et al., 2006a*). Previous studies of Caucasians have also reported high levels of variation explained by BMI: 84.1% (*Tovée et al., 2006*), 83.3% (*Swami et al., 2006a*), 76.8% (*Swami & Tovée, 2007*),73.7% (*Tovee & Cornelissen, 1999*), 73.5% (*Tovee et al., 1998*), 70.3 to 73.3% (*Swami & Tovée, 2007*), although using 3D rotating images the impact of BMI was lower at 53% (*Smith, Cornelissen & Tovée, 2007*). This range of values for 2D images covers the data reported here. Consistent with our work, previous multiple regression analyses of rated attractiveness of the female body using digitally manufactured stimuli suggested that BMI was twice as important as WHR for the rated attractiveness in Poland (*Kościński, 2012*; *Kościński, 2013*; *Kościński, 2014*).

Few previous studies of the impact of body fatness and WHR have been conducted in African populations. In contrast to our data, 82.5% of the variance in attractiveness was explained by BMI and only 7.5% by WHR in South African Zulus (*Tovée et al., 2006*), which suggested African populations do not differ from Caucasians and Asians. In contrast, we found much lower % explained variation by body fat and a much greater role for WHR in all four of the African populations in our sample. In fact, in the sample from Nigeria, WHR was more significant than BMI in the multiple regression analyses. The greater role for WHR in the four African populations is consistent with previous studies that have suggested a preference for a more extreme low WHR in African populations (*Furnham, Moutafi & Baguma, 2002*) and a preference for a lower WHR among African Americans compared to US Caucasians (*Freedman et al., 2007*; *Freedman et al., 2004*). The reason why WHR may play a greater role in African populations is presently unclear. One potential factor maybe the role of the buttocks in assessments of physical attractiveness among Africans (*Marlowe, Apicella & Reed, 2005*) and African

Americans (*Cunningham et al., 1995*). This ethnic difference is apparent in the differences in ethnic ideals with respect to buttock augmentation surgery (*Roberts et al., 2006*) in which Asians prefer very small and African Americans very large buttocks. This difference may accentuate the importance of WHR in attractiveness ratings by African populations.

In addition, there are ethnic differences in the reported consequences of obesity for various health related parameters. For example, obese African Americans show much greater risk of developing insulin resistance and diabetes than either Caucasians or Hispanics, as they become obese, but Hispanics show a much greater risk of hepatic steatosis (*Speakman & Goran, 2010*). It is potentially the case that WHR provides a much better signal of health in Africans than in other populations.

## Evolutionary aspects and optimal BMI

Many previous studies have set the observations that raters prefer women with given WHRs and BMIs into a *post hoc* evolutionary context by suggesting that BMI and WHR are honest signals of both health and fertility (*Bigaard et al., 2004*; *Singh, 1995a*; *Wass et al., 1997*). Few studies, however, have attempted to rigorously test this suggestion by comparing the actual pattern of variation in attractiveness, as a function of fatness or WHR, to that expected *a priori* on the basis of epidemiological data on the relationships between fatness, health and fertility. We attempted to fill this gap by constructing a mathematical model relating fatness to future mortality risk (incorporating both health and fertility effects) using data from several large epidemiological studies that have related BMI to all cause mortality in Caucasians (*Whitlock et al., 2009*), Asians (*Zheng et al., 2011*) and African Americans (*Cohen et al., 2014*; *Cohen et al., 2012*; *Flegal et al., 2013*) and BMI at age 20 to future reproductive success in over 33,000 females Caucasians (*Jacobsen et al., 2013*). Factoring these two effects into the model suggested the optimal female BMI should be around 22.4 to 23.2. A limitation of our study is that we could not find data on the link of mortality to BMI for Africans living in Africa, for which we substituted data on African Americans, and we could only find fecundity data based on a large sample size for Caucasians. The similarity of the mortality patterns between Caucasians, Asians and African Americans (Fig. 1A), however, lends some confidence that ethnic effects on these relationships are relatively small and the derived optima are probably appropriate for the populations we studied. An additional potential factor is that body fat stores may provide a resource base to ensure survival during periods of famine (the thrifty gene hypothesis). Using a previous mathematical model relating famine survival to fat storage (*Speakman & Westerterp, 2013*) we predicted that if famine mitigation was also important the optimum BMI might rise slightly to between 24.0 and 24.8 but the shape of the curve relating attractiveness to body fatness would be more steeply negative at lower levels of body fatness (Fig. 1D).

This predicted peak relationship between body fatness and physical attractiveness with a maximum attractiveness around a BMI of 22.4 to 24.8 was not supported by the data we collected in any population. Over the range of BMIs that we studied (19 to 40) there was a negative linear between attractiveness and BMI (Fig. S2). This range was adequate

to detect a peak, if such existed, at the position we predicted. In none of the populations over this BMI range was there any indication of a peak in the relationship (as judged by the significance of adding additional polynomial terms to the regression model). Similar to our data, a linear negative relationship between BMI and physical attractiveness was observed previously over the BMI range 18 to 26 (*Swami & Tovée, 2007*), and a linear negative relationship was observed between attractiveness and body fatness over a range from 20 to 35% body fat (*Smith, Cornelissen & Tovée, 2007*). Using the same image set as used here *Faries & Bartholomew (2012)* also reported a linear negative relationship between attractiveness and BMI rated by US college students of mixed ethnicity.

Hence, if there was a peak physical attractiveness, in all ten of the populations we studied the peak was at least as low as BMI = 19 and potentially lower. This was consequently at least 3.5 BMI units below the predictions of the evolutionary model. For an average height woman (1.55m) the difference between the predicted and observed maximum was at least 10 kg of body weight. This is an enormous difference in body weight and based on these data we can clearly reject the evolutionary models, as formulated, based on health, fertility and famine survival. We did not have more extreme body compositions included into the images presented to the raters, and hence there might be a maximum attractiveness at a lower BMI than the lowest BMI in our image set. In fact data from previous studies suggest that there may be a peak in the relationship between BMI and attractiveness (*Swami et al., 2006a*; *Swami & Tovée, 2007*; *Tovée et al., 2006*; *Tovee et al., 1999*; *Tovee et al., 1998*) at a BMI around 18 to 20. In all these cases the authors fitted a 3 term polynomial with WHR as an additional term, but in only one study were actual coefficients reported in the publication. Hence it was not possible to explicitly solve the equations to locate the peak. We consequently recoded the data from the plots presented in the figures (using the software package Data-thief) and fitted our own curves to the data and then solved these curves for the maxima by differentiating them and solving them for $f'(x) = 0$ (Table 4). This reanalysis of previously published data clearly shows that for most populations the maximal physical attractiveness occurred at a BMI between 18.4 and 21.4 (mean of all studies excluding 2b and 4b in Table 4 = 20.152, sd = 1.012, $n = 7$). The two excluded studies are discussed below. This mean peak attractiveness sits 2.4 to 4.6 BMI units below the prediction (about 9 to 16.5 kg for an average height woman). This estimated peak attractiveness at a BMI at 18.4 to 21.4 is consistent with many other data. For example, the BMIs of Playboy centerfolds and glamour models over the last 50 years are almost all in the range 17 to 20 (*Katzmarzyk & Davis, 2001*; *Tovee et al., 1999*; *Voracek & Fisher, 2002*). Women and men asked to manipulate female 3D computer models to make them maximally attractive make them have BMIs of 18.9 and 18.8 respectively (*Crossley, Cornelissen & Tovee, 2012*). The biggest outlier in previous studies of attractiveness at low BMI was the observation that in Poland the highest rated attractiveness was at a BMI of 15 (*Kościński, 2013*), and potentially lower as this was the smallest stimulus in the set presented.

**Table 4** Parameters of 3rd order polynomials fitted to data on attractiveness as a function of BMI in previous studies in the literature, along with the estimated BMI at 'peak' attractiveness obtained by differentiating the fitted curves and solving the resultant quadratic equations for $f(x) = 0$ in the range 30 10.

| Study | $x^3$ | $x^2$ | $x$ | Constant | $r^2$ | Peak |
|---|---|---|---|---|---|---|
| 1 | 0.0019 | −0.1521 | 3.8397 | −26.003 | 0.732 | 20.486 |
| 2a | 0.0016 | −0.1421 | 3.7809 | −26.817 | 0.842 | 20.185 |
| 2b | 0.0007 | −0.0651 | 2.0092 | −14.419 | 0.837 | 28.941 |
| 3a | 0.0013 | −0.1053 | 2.5562 | −15.628 | 0.784 | 18.423 |
| 3b | 0.0016 | −0.1369 | 3.6632 | −26.131 | 0.830 | 21.431 |
| 4a | 0.0023 | −0.1953 | 5.1321 | −36.786 | 0.826 | 20.731 |
| 4b | 0.0007 | −0.0654 | 1.973 | −12.878 | 0.800 | 25.633 |
| 4c | 0.0021 | −0.1789 | 4.6963 | −33.44 | 0.768 | 20.591 |
| 5 | 0.0018 | −0.1503 | 3.7823 | −25.23 | 0.725 | 19.215 |

**Notes.**

Studies were (1) (*Tovee et al., 1999*), (2) (*Tovée et al., 2006*) (a) British (b) Zulus, (3) (*Swami et al., 2010*), (a) Japanese (b) British, (4) *Swami & Tovée (2007)* (a) British (b) Hill tribe Thai (c) city Thai, (5) (*Tovee et al., 1998*)

Why do the data for these modern societies seem to deviate so widely from the evolutionary model predictions about the most attractive level of body fatness? One potential interpretation is that the populations studied here are all exposed to the same western media which promotes a thin female body ideal (*Groesz, Levine & Murnen, 2002*; *Posavac, Posavac & Posavac, 1998*). It is difficult however to separate cause and effect. Does media exposure drive people's perceptions of attractiveness? Or is the 'thin ideal' in the media simply reflecting what people already see as attractive? The fact the populations differed significantly in their perceptions of the importance of WHR suggests that in fact their opinions are not all homogenized by exposure to the same western media images of what is attractive. The data were not consistent with the suggestion that people are attracted to averageness in their own population (*Kościński, 2012*) since the universal preference for low BMI contrasted the much higher and more variable levels of average BMI among the rating populations (Table 1).

A potential problem with studies such as ours, and all previous studies of the role of BMI or body fatness, based on 2D images or 3D models, is that the people making the ratings are given no instructions about the age of the subjects. Body fatness and BMI are both strongly related to age (*Speakman & Westerterp, 2010*) as to a lesser extent is WHR (*Marlowe, Apicella & Reed, 2005*). Hence, individuals rating the images may be using BMI as a proxy to estimate the age of the subjects. Our observers were definitely sensitive to the ages of the individuals in the pictures, despite there being no immediately obvious way they could tell their ages. Were they also using BMI as a cue to the age of the subjects? There was some evidence to support this hypothesis. When individuals matched up the models ages to their pictures, there was a strong association between the estimated age and both BF% and BMI but not to their actual ages (Fig. 5). This suggests that people viewing the images used body fatness to estimate the age of the subjects. In the evolutionary model of the impact of fatness and fertility we assumed that age and BMI were independent.

However, fertility is strongly dependent on age, in part because of the declining ovarian reserve as a function of age (*Wallace & Kelsey, 2010*). However, fertility reaches a peak in the late teens and early 20s because prior to age 20 there is an increased risk of annovulatory cycles. The relationship between infertility and age based on literature data for Caucasians (*Henry, 1961*; *Leridon, 1978*; *Leridon, 2008*; *Menken & Larsen, 1986*; *Pittenger, 1973*; *Trussell & Wilson, 1985*; *Vincent, 1950*) is shown in Fig. S3 and a 4th order polynomial explained 98.3% of the variation in infertility. The best fit equation was

$$y_5 = 0.0052A^4 - 0.6164A^3 + 27.105A^2 - 514.95A + 3558.1 \qquad (12)$$

where $y_5$ is the age related infertility per thousand population, and A is the age. Given the relationship between BMI and estimated age of the subjects (Fig. 5) we can use the derived fitted relationship

$$A = 0.027x^2 + 3.1717x - 30.599 \qquad (13)$$

where $x$ is the BMI, to derive the expected relationship between BMI and mortality risk if BMI is used only as a proxy for age. Substituting Eq. (13) into Eq. (12) gives

$$y_5 = 0.0052(0.027x^2 + 3.1717x - 30.599)^4 - 0.6164(0.027x^2 + 3.1717x - 30.599)^3$$
$$+ 27.105(0.027x^2 + 3.1717x - 30.599)^2 - 996.68x + 12495.57 \qquad (14)$$

Differentiating Eq. (14) gives

$$\frac{dy}{dx} = 0.0208(0.027x^2 + 3.1717x - 30.599)^3 - 1.8492(0.027x^2 + 3.1717x - 30.599)^2$$
$$+ 1.4637x^2 + 171.94x - 2655.45$$

Expanding the brackets and collecting terms gives the 6th order polynomial

$$\frac{dy}{dx} = 0.000000409x^6 + 0.0001445x^5 + 0.01421x^4 + 0.019753x^3 - 31.793x^2$$
$$+ 716.18x - 3790.53. \qquad (15)$$

And solving Eq. (15) for $f(x) = 0$ gives a single root in the range 16 to 50 at $x = 17.41$. Although, this is significantly lower than the mean peak attractiveness of 20.15 across the studies in Table 4 (one sample $t$-test $= 7.17$, $p < 0.001$), the local minimum at 17.41 is very shallow and there is very little difference over the range from 16 to 21, which encompasses most of the maxima in attractiveness in the studies summarized in Table 4. This analysis suggests that the shape of the relationship between BMI and physical attractiveness may come about primarily because subjects in such experiments use BMI as an indicator of subject age, and then attractiveness is primarily gauged on the evolutionary significance of the estimated age. The strong link of age to fertility results in subjects rating the pictures with BMIs around 20 as most attractive because they would be aged 19 to 22 and hence most fertile. Additional factors such as health relationships to BMI and the role of famine, and indeed the effects of BMI at a fixed age on fertility appear negligible by comparison.

There were 2 exceptions to this pattern of a peak in the range 18 to 19 (Table 4). The hill tribes people of northern Thailand had a maximum attractiveness at BMI = 25.6 (*Swami & Tovée, 2007*), and the Zulus of South Africa had a maximum attractiveness at BMI = 28.9. In these latter cases the maximum clearly sits much closer to the predictions of the evolutionary model derived here. Although insufficient images and data were available to fit an exact curve it seems likely that similar data with higher BMIs at maximal levels of attractiveness would also be observed among the Hadza of Tanzania (*Marlowe & Wetsman, 2001*).One hypothesis is that these divergent patterns emerge because these communities do not use BMI as a proxy for age, and the resultant pattern then matches more closely the evolutionary predictions from the model excluding such a link. This may be because in these communities that are all resource poor, body fatness does not increase with age in the same way it does in modern societies (*Lawrence et al., 1987*; *Prentice et al., 1981*), and hence low BMI is not an honest signal of youthfulness.

## CONCLUSIONS

Our data confirm previous studies showing body fatness (and BMI) explained more of the variation in ratings of physical attractiveness than waist to hip ratio (WHR). Novel here was the demonstration that WHR played a more important role (greater $r^2$) in African than in Asian populations. The relationships between attractiveness and body fatness (BMI) did not match the predictions from a theoretical model based on large epidemiological studies of the impacts of body fatness on health and fertility, combined with the relationship between fatness and famine survival. An explanation for this discrepancy is that raters used the body fatness of the subject images as a proxy for subject age, and age is more strongly linked to fertility than is BMI (independent of age). A model based on this assumption matched our, and previous, data relating fatness to attractiveness. Overall our data and modeling suggest that the role of BMI in ratings of attractiveness may have been overstated because of the covariance of BMI with age. Future studies aiming to quantify the contribution of body fatness (BMI or BF%) and WHR to ratings of physical attractiveness need to remove any covariance between fatness (BMI or BF%), WHR and age.

## ACKNOWLEDGEMENTS

We are grateful to all the participants from all the countries and all the members of Molecular Energetics Group for their help on the investigation and discussion of the results.

### Funding

This work was supported by NSFC grant 91431102 from the National Science Foundation of China. The funders had no role in study design, data collection and analysis, decision to publish, or preparation of the manuscript.

## Grant Disclosures

The following grant information was disclosed by the authors:
NSFC: 91431102.
National Science Foundation of China.

## Competing Interests

The authors declare there are no competing interests.

## Author Contributions

- Guanlin Wang performed the experiments, analyzed the data, wrote the paper, prepared figures and/or tables, reviewed drafts of the paper.
- Kurosh Djafarian, Chima A. Egedigwe, Asmaa El Hamdouchi, Robert Ojiambo, Harris Ramuth, Sandra Johanna Wallner-Liebmann, Sonja Lackner, Adama Diouf, Justina Sauciuvenaite, Catherine Hambly and Lobke M. Vaanholt performed the experiments.
- Mark D. Faries contributed reagents/materials/analysis tools, contributed to the study design and commented on the draft of this paper.
- John R. Speakman conceived and designed the experiments, wrote the paper, prepared figures and/or tables, and built the evolutionary model.

## Human Ethics

The following information was supplied relating to ethical approvals (i.e., approving body and any reference numbers):

All the procedures for the overall study were ethically reviewed and approved by the Chinese Academy of Sciences, Institute of Genetics and Developmental Biology Institutional Review Board (IGDB-2013-IRB-005). In addition, local ethical approval was also obtained at the UK site from the University of Aberdeen College of Life Science and Medicine Ethical Review Board (CERB/2014/12/1123).

## Supplemental Information

Supplemental information for this article can be found online at http://dx.doi.org/10.7717/peerj.1155#supplemental-information.

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
