# Peer review of "The relationship of female physical attractiveness to body fatness"

_PeerJ, doi:10.7717/peerj.1155_

## Round 0.1 · original submission · Minor Revisions

Dear Author's. As you can see in the attached reviews all three reviewers felt only minor revisions were needed for publication. These should all be able to be addressed easily I believe (and some are just grammatical). Please address these in a separate word document tothe revised manuscript and give special attention to reviewer 3's comments on the validity of the findings.

Thank you for your submission and I look forward to seeing this interesting work published very soon.

Reviewer 1 ·

Basic reporting

There are a few grammar issues that should be revised. See specific comments to the authors below.

Experimental design

No comments regarding experimental design.

Validity of the findings

No comments regarding validity of findings.

Additional comments

Review of ‘The relationship of body fatness to female attractiveness’

This is an interesting and relatively straightforward study that examines the link between attractiveness and body fatness and waist-to-hip ratio (WHR). Consistent with previous studies, the authors find that body fatness explains more variation in physical attractiveness than WHR. The authors additionally find that the relative importance of body fatness and WHR in attractiveness was dependent on culture. Interestingly, the authors use a mathematical model to predict the optimally-attractive BMI; however, the prediction of the model that BMIs of 22.4-24.8 should be most attractive was not supported by the data collected, and instead, the observed peak BMI was lower. I have only relative minor concerns with the manuscript, and these concerns are detailed below:

P5, L 5: ‘aspects such economic parameters’? should this be ‘such as’?

P5, L 15: significant factors in what? I suggest rephrasing ‘…significant factors in mate choice’.

P5, L 16: ‘relative importance of these different dimensions’: relative important in what? mate choice? rephrase to be explicit.

P5, L 21: ‘around 0.7 were been generally interpreted’; rephrase, as this doesn’t make sense as written.

P5, L 20 – P6, L2: As written, the logic of this is unclear. WHR of 0.7 is a ‘marker of female fertility and health, since high values of WHR are related to elevated risk of cardiovascular disease’. are values higher than 0.7 correlated with these diseases? if so, reword to make this clear. also, ‘since’ should be changed to ‘because’.

P6, L 2-3: This could be more concise; e.g. delete the first part of this sentence and rephrase to simply state that ‘Lassek and Gaulin have suggested that WHR is not…’

P6, L 10: delete ‘and many other studies’, as it doesn’t fit at the end of this sentence and simply add ‘e.g.’ before your parenthetical list of citations.

P18, L21-22: Interesting result.

P20 (1st page of Discussion), L 21: comma should be replaced with a period after ‘…2002)’

Reviewer 2 ·

Basic reporting

The manuscript is not poorly written but does contain grammatical errors that should be addressed. There is an ample amount of run-on sentences that could be broken up into several sentences for a smoother, easier read. Also, there are a few spelling errors. For example, the second sentence of the introduction is long and could be broken into perhaps two sentences. In summary, the manuscript needs more proofreading and correcting in regards to grammar.

The graphs in figure 1 does not have labeled x axes, which makes it difficult to corroborate the figure with the appropriate results.

In the abstract, the initials BF% are not defined. It is made clear later in the manuscript that BF% stands for body fat percentage but that should be stated in the abstract as well.

It may be useful to add, if possible, a brief statement at the end of the introduction that states what the results are and what they may indicate. This way the reader can get a quick overview of the finding the manuscript will be contributing to the scientific community.

Experimental design

The last sentence of the "evolutionary model" paragraph appears to state a hypothesis. Perhaps this hypothesis, if it is a hypothesis, should also appear somewhere toward the end of the introduction.

The "female body images" section states that the 21 images that were utilized were used to evaluate the role of fatness in the perception of physical attractiveness in U.S. college students. However, all of the participants come from ten countries, none of which includes the U.S. So, it appears that the study is really evaluating the the role of fatness in the perception of physical attractiveness in participants from those ten countries.

The "female body images" section states that female images were "specifically selected to break any correlation of BF% to WHR". How was this achieved?

Why was the age of the subjects not controlled for?

How were participants recruited?

The manuscript addresses how the participants' age was used in the study, but what was the purpose of reporting the participants' ethnicity, height, and weight?

When the participants judged the 21 images in regards to attractiveness were all the participants in one room together? Did they do the judging in separate rooms at the same time? Did they do it one after the other?

Toward the end of the "procedure" section there is mention of applying a different protocol to the U.K. subjects. Why not apply that protocol to all the other subjects as well? Doing that may make for a more solid basis for comparing the outcomes of that protocol with the one used in the study.

When participants matched the 21 images with ages, were they told to match between the ages of 18 and 50 or were they just guessing the ages?

Perhaps describe briefly what all cause mortality is.

Throughout the manuscript the word "peaked" is used to describe relationships, functions, etc. What exactly does that word mean?

It is sated that WHR is an independent factor. What exactly does that mean?

Validity of the findings

I agree with the conclusions drawn from the results. I gathered that BMI or body fatness is a stronger indicator of physical attractiveness than WHR, but I am not certain what is meant when the authors say that body fatness explains the variation in physical attractiveness more so than WHR. Is that synonymous with saying that BMI or body fatness is a stronger indicator of physical attractiveness than WHR?

Additional comments

I found the work interesting and eye catching.

·

Basic reporting

Dear editor, Wang et al. describe here the results of an extensive study on the relationship between female fatness and their attractiveness. Many studies of this kind already exist. However, authors offer new data of attractiveness from 10 different populations, a nice synthesis on the relationship between Body Mass Index (BMI) and health, as well as a nice discussion of the relationship between BMI and perceived age. Moreover, authors did not overstate their results, nor they slipped into evolutionary nonsensical interpretation of their findings. Therefore, I see this paper as a nice contribution to attractiveness literature, that deserve to be published (provided they address a issue about the interpretation of their data, cf below).

Experimental design

The experimental design is simple but effective.

Validity of the findings

All data have been carefully examined. However, I am afraid that I do not share the interpretation of two main findings (minor comments are provided in the General Comments for Authors):

1. About the lack of gender effect on attractiveness. Most studies performed on Caucasian report little sex differences in the assessment of attractiveness, but often gender effects are found in non Caucasian samples. Here authors argue that they did not find any gender effect. Yet, I think this conclusion is the result of applying a very conservative correction for multiple testing (Bonferroni) applied across all countries indiscriminately. Such procedure seriously impede the statistical power of the analysis. In fact, looking at p-values before the correction for multiple testing, authors found 17 (if I am counting well) significant p-values out of 188 (=189 -1 for the NA), while around 9 would be expected by chance. In particular, among the so called African populations, 12 results out of 63 (=21*3) are significant, which significantly differ from 5% expectation (binomial test, p=0.03). While there is nothing wrong in what the authors have done, I would acknowledge that gender differences may be present, especially in some populations (Iran, Kenya, Nigeria), and I would indicate whether considering males only for all populations (and thereby excluding Senegal) does influence qualitatively the conclusion of this paper or not. I believe it will not, but it would be nice to show so. I am not convinced by the argument saying that for evolutionary reasons males and females should judge attractiveness in the same way. Within sex competition takes different forms in both sexes and this could bias perception of attractiveness.

2. About the negative relationship between fatness and attractiveness. Authors argue that in there dataset slimmest females are perceived as more attractive. Yet, on figure 3 and figure S2, it is clear that the pictures with the highest attractiveness score are not the slimmest at all, but those lying between 20 and 25 kg.m^-2. Moreover, the slimmest always get bad ratings and because stimuli are generated from many pictures that may not due to something else than BMI. While I agree that the trends are not clearly quadratic, there are not linear either! Using linear regression is fine but only to a point. If I wanted to be picky I would even say it is not the right tool because assumptions to run linear regressions are not met here (e.g. residuals are not normally distributed). Because there is no simple alternative, I would stick with these regressions but I would however discussed why some particular body mass seem to deviate a lot from the regression line. I would actually argue that the highest ranked stimuli as a BMI falling within (or close to) the range of peaks reported in table 4. Perhaps preferences are not as off from the predictions authors make as they claim. The more fundamental problem here is the shape of the relationship one expect for preferences. We way linear or quadratic mainly because we do not know, but many other preference function may exist.

Additional comments

Good work, see my comments above.
Here, I will just put some minor comments.

- Abstract: calling your polynomial fit “mathematical model” is a bit misleading as this is not a theoretical model but a statistical one, I would call this statistical model. Considering Iran and Mauritius as Asian countries is perhaps not ideal. In particular, in Geohive, which the author used Mauritius is classified as Eastern Africa.

- Introduction: fine.

- Methods: you say that body fat is advantageous after famine but you build that on an hypothesis only. Maybe provide citation of analysis of real dataset instead. Mention here where you got data about relationship between survival and fecundity and attractiveness. Introduce in few words the x-ray absorptionmetry method. What you called “multivariate analyses” are called “multiple regressions” and should be distinguished from “mutlivariate regression”, so change your names to avoid confusion. In statistics, “multivariate” means multiple outcomes (y – variables), not multiple predictors.

- Results: Eqn 1.1 say that x stands for BMI. Page 13 line 20, you say fig 1.B provide data pooled over 20 years, but you show annual rate instead on this figure. Page 14 line 2: “Best Fit”: according to which criterion do you say a fit is best (LRT?) ? Page 14 line 19: justify why you sum variables and not multiple them (or sum with weight or other...). It is a big assumption and probably the best because you have not much information to do otherwise but it should be explicit. Page 15 line 21, it is not just lowest mortality since you included fecundity in the summation. Page 16 line 4: polynomial of degree 3 are not symmetrical. Page 16 line 17: again, why summing? Justify. Page 18 line 9: explain by how many test you corrected during the Bonferroni procedure. Page 18 line 16: false statement: image with lower BF were not always rated higher, just look at your figures. Page 19 line 5: again this is not a multivariate analysis. Page 19 line 13: “(P < .05 > 0.01)” is not a very clear notation.

- Discussion: page 20 line 13: I disagree that males and females should perceive attractiveness the same. I guess you will fail to find one theoretical paper on mate choice predicting that. I would remove this argument. Page 25 line 15: why do you speak about exponential fit here, there are none on fig S2. Page 26 line 6 and page 27 line 1: I do not obtain same number of kilograms than you. Example: 10/(1.55^2) is not 3.5 but 4.1... Page 29 line 6: equation is not the same as on the plot it seems...Page 31 line 8: yes and no, that age may underline the reason why low BMI is attractive does not make it less attractive. Ultimately there are always reasons behind the preference of something. So don't say it has been overstated.
- Fig 1. A: what are curves, you say it for B. but not for A.
- Fig 1. B. C. In y axis you wrote mortality but it is not that since fecundity is accounted for.
- Table 3: again do not call that multivariate. For non-significant P-values, report the actual value, not just P>0.05!

---

## Round 0.2 · accepted · Accept

The revisions have been incorporated and have adequately addressed the concerns.